# Subclinical Atherosclerosis Imaging in People Living with HIV

**DOI:** 10.3390/jcm8081125

**Published:** 2019-07-29

**Authors:** Isabella C. Schoepf, Ronny R. Buechel, Helen Kovari, Dima A. Hammoud, Philip E. Tarr

**Affiliations:** 1University Department of Medicine and Infectious Diseases Service, Kantonsspital Baselland, University of Basel, 4101 Bruderholz, Switzerland; 2Department of Nuclear Medicine, Cardiac Imaging, University Hospital Zurich, University of Zurich, 8091 Zurich, Switzerland; 3Division of Infectious Diseases and Hospital Epidemiology, University of Zurich, 8091 Zurich, Switzerland; 4Center for Infectious Disease Imaging, Radiology and Imaging Sciences, National Institutes of Health, Bethesda, MD 20892, USA

**Keywords:** subclinical coronary artery disease, accelerated atherosclerosis, HIV infection, carotid intima-media thickness, coronary calcium scoring, coronary CT angiography, magnetic resonance angiography, fluorodeoxyglucose positron emission tomography

## Abstract

In many, but not all studies, people living with HIV (PLWH) have an increased risk of coronary artery disease (CAD) events compared to the general population. This has generated considerable interest in the early, non-invasive detection of asymptomatic (subclinical) atherosclerosis in PLWH. Ultrasound studies assessing carotid artery intima-media thickness (CIMT) have tended to show a somewhat greater thickness in HIV+ compared to HIV−, likely due to an increased prevalence of cardiovascular (CV) risk factors in PLWH. Coronary artery calcification (CAC) determination by non-contrast computed tomography (CT) seems promising to predict CV events but is limited to the detection of calcified plaque. Coronary CT angiography (CCTA) detects calcified and non-calcified plaque and predicts CAD better than either CAC or CIMT. A normal CCTA predicts survival free of CV events over a very long time-span. Research imaging techniques, including black-blood magnetic resonance imaging of the vessel wall and 18F-fluorodeoxyglucose positron emission tomography for the assessment of arterial inflammation have provided insights into the prevalence of HIV-vasculopathy and associated risk factors, but their clinical applicability remains limited. Therefore, CCTA currently appears as the most promising cardiac imaging modality in PLWH for the evaluation of suspected CAD, particularly in patients <50 years, in whom most atherosclerotic coronary lesions are non-calcified.

## 1. Introduction

Cardiovascular disease (CVD) has become one of the leading causes of death in people living with HIV (PLWH) worldwide [1,2,3,4]. Reducing the burden of CVD, particularly coronary artery disease (CAD), is therefore emerging as a major public health goal in medical care for PLWH [5,6].

Data is inconsistent with respect to whether PLWH have an increased incidence of CAD events compared to the general population [2,7] or not [8,9,10]. Early reports have observed a 2- to almost 4-fold increased rate in CAD events as compared to HIV-negative patients [11,12], but traditional risk factors, particularly smoking, and socioeconomic factors were not always adjusted for in those studies [13]. Danish data suggest a 1.5- to 2-fold increase in CAD events in PLWH vs. controls [14]; a 2-fold CAD event increase in PLWH was also the conclusion of a recent systematic review and meta-analysis [2].

Other studies have reported smaller increases [15,16,17] in cardiovascular (CV) events in PLWH. In a large cohort study from California that evaluated the myocardial infarction (MI) risk from 1996 to 2011, a declining relative risk for MI was observed in PLWH vs. HIV-negative persons over time. Between 2010 and 2011, no increased risk of MI was seen in PLWH [17]. In a cohort of US veterans, a 50% increased risk in acute MI was recorded, after adjusting for Framingham risk score, comorbidities and substance use [16]. In another large North American analysis, HIV infection was associated with only a 21% increased incidence of CV events in comparison with a well-matched HIV-negative control group, and the risk increase was not statistically significant in the age group ≥60 years of age [15]. Moreover, in recent studies from Switzerland and Denmark, no increased risk of CAD events and, in Switzerland, no increased subclinical atherosclerosis prevalence was described, perhaps due to the prevalence of modern antiretroviral therapy (ART) regimens, reliably suppressive ART, decreased smoking prevalence, and access to regular medical follow-up [8,9,18]. Also, the data do not generally suggest that CV events occur earlier in PLWH, contradicting the widely held notion of “accelerated atherosclerosis” or even “accelerated aging”. Some reports that CV events occur at a younger age seem to be related to the younger median age of many HIV+ populations compared to the general population [19].

The pathogenesis of CV events in HIV-positive individuals likely represents a complex interaction of different factors that contribute to the development of atherosclerosis [18,20,21,22,23]. Studies in multiple countries have consistently recorded an increased prevalence of smoking and drug use in PLWH [11,13]. A link between inflammation related to HIV infection and CAD in PLWH is well documented. Several serum biomarkers that are associated with CV risk or overall mortality may be elevated in PLWH compared to HIV-negative persons, including biomarkers of systemic inflammation (high sensitivity C-reactive protein, interleukin-6, soluble tumor necrosis factor-α receptor 1) [24,25,26,27], coagulation (D-dimer, fibrinogen) [24,25,26,27], monocyte activation (lipopolysaccharide, soluble CD14, CD163, CC-chemokine ligand 2) [24,28,29,30,31], and endothelial dysfunction (intercellular adhesion molecule 1) [24]. Although the exact mechanisms underlying the development of atherosclerosis in PLWH have not been conclusively determined, smooth muscle cell proliferation may play an important role [32]. Moreover, endothelial dysfunction due to adhesion molecules may lead to adhesion of circulating leukocytes, transmigration of monocytes/macrophages to the intima and subsequently to vascular inflammation [20,32,33]. In combination with procoagulatory mechanisms in the setting of increased concentrations of D-dimer and coagulation factors, and in the presence of other metabolic disorders (e.g., dyslipidemia), the progression of atherosclerosis might be enhanced in PLWH [20,32].

It is well accepted that viral suppression, in the setting of adequate ART, is associated with lower levels of inflammatory biomarkers and decreased CV risk [34,35]. Advanced immunosuppression has contributed to CV risk in PLWH in some [15,36,37] but not in other studies [38,39]. 

Also, certain ART agents have been associated with increased CV risk. This notion is most reliably based on findings from the large, multinational, observational D:A:D study [20,23,39]. Dyslipidemia was among the first described deleterious metabolic effects of ART, most notably in the setting of certain protease inhibitors, i.e., lopinavir, indinavir and darunavir [23,40,41]. Within the drug class of nucleoside reverse transcriptase inhibitors, abacavir and didanosine were found to increase CV risk [42].

The potentially increased CV event rate in PLWH has generated considerable interest in early detection of asymptomatic (subclinical) atherosclerosis, in order to improve CAD event prediction and to allow appropriate CVD prevention by early intervention [21]. Most published literature on noninvasive cardiac imaging for the detection of subclinical atherosclerosis in PLWH has focused on carotid artery intima-media thickness (CIMT), coronary artery calcification (CAC), coronary CT angiography (CCTA) [18,43,44] and, more recently, magnetic resonance imaging of the vessel wall (MRI) and 18F- fluorodeoxyglucose positron emission tomography (FDG-PET).

## 2. Carotid Artery Intima-Media Thickness (CIMT)

CIMT is an ultrasound technique to measure the thickness of the two layers of the carotid artery wall, the intima and the media [45]. CIMT is well recorded to predict future CV events in the general population [46,47,48,49]. According to the results of a 2009 meta-analysis that analyzed six cross-sectional, seven case-control and 13 cohort studies (5456 HIV positive and 3600 HIV negative patients) PLWH tend to show a greater thickness in CIMT (0.04 mm thicker; 95% confidence interval: 0.02–0.06 mm, *p* < 0.001) when compared to HIV-negative controls [50]. However, the findings are not concordant with some studies showing increased CIMT was increased in PLWH compared to HIV-negative controls [12,51,52,53,54,55,56,57,58,59,60,61,62,63,64,65,66,67,68,69,70,71] while other studies did not show a difference or showed only weakly increased CIMT in PLWH [12,45,50,51,52,53,54,55,56,57,58,59,61,62,63,64,65,66,67,68]. Those discordant findings may be attributable to differences in study design, participant characteristics, duration of follow-up, and different approaches of ultrasound measurement [45,50].

The largest differences in CIMT between HIV-positive and HIV-negative participants were noted in studies with the greatest demographic differences between the analyzed groups [45,50]. Moreover, small studies were more likely than larger studies to identify an increase in CIMT in PLWH vs. controls [45,50]. In addition, a large report of repeated CIMT measurements over a median of 7 years did not find accelerated CIMT progression in PLWH (747 women, 530 men) compared to HIV-negative controls (264 women, 284 men), but focal plaque prevalence was increased, after adjusting for traditional CV risk factors [70]. These findings are in accordance with another report that observed no different progression in CIMT over 144 weeks in 133 extensively matched HIV+ and HIV− participants [59].

CIMT has a number of limitations for the prediction of CV events, including a limited correlation with angiographically defined atherosclerosis [72,73], limited improvement in CV event prediction by the addition of CIMT to the Framingham risk score [46], and different results when CIMT findings at the common carotid artery level are compared with results obtained at the carotid bifurcation and/or the internal carotid artery level [45,61,65]. Finally, CIMT measurement is dependent on investigator experience, with the reproducibility of results generally being higher in research settings than in practitioner-based settings [74].

## 3. Coronary Artery Calcium (CAC) Scoring

CAC scoring using non-contrast enhanced CT is a well-established and easily applicable tool for detection and quantification of coronary calcifications [75,76,77,78,79,80] (Figure 1). Applying different scoring systems, most frequently the Agatston score, based on the landmark work of Arthur Agatston in the late 1980s [81], CAC scoring has emerged as a robust non-invasive atherosclerosis imaging modality characterized by high inter- and intra-observer reliability [74,80,82].

In the general population, there is a strong correlation between CAC score and future CV endpoints [74,80,83,84,85,86]. Persons with no detectable coronary calcium have a very low risk for CV events over the following years [87,88] and a ten-year survival of 99.4% [89]. Longitudinal CAC studies have suggested that annual CAC score change of ≥15% may constitute CVD progression [90]. CV event prediction by CAC and CIMT were similar in one report [91], but CAC was a more reliable CV event predictor than CIMT in several other reports [86,92,93,94]. CV event prediction may be improved when CAC is added to Framingham risk score [88,89].

Current evidence remains equivocal as to whether the presence of HIV is associated with an increased prevalence of coronary calcifications [18,71,95,96]. An increased vascular age was identified in Italian PLWH compared with age-specific CAC percentiles based on the MESA study done in a general US population [95]. However, these findings were not confirmed in a recent study assessing a large Swiss cohort [18]; other studies have also found similar CAC scores in HIV-positive and HIV-negative persons [18,50,96]. High pre-treatment HIV viremia levels were associated with a CAC score >0 [18]. In the US MACS study, CAC scores were elevated in those with metabolic syndrome but were not altered by HIV serostatus [57,96].

A major limitation of CAC is the inability to detect non-calcified plaque. This is of importance because a non-calcified or only partially calcified plaque is more likely to take on morphological features typical of high-risk plaque which is prone to adverse future CV events (e.g., plaque rupture or erosion potentially leading to MI) [97,98]. In addition, coronary artery plaque is predominantly non-calcified in patients <50 years of age [80,99,100]. This point may be particularly relevant to PLWH: as, in Western countries, even though HIV-positive populations are aging, their median age is still relatively low, e.g., 48 years in the Swiss HIV cohort study (www.shcs.ch and [101]). Their Framingham risk scores and CAD event rates have also been relatively low, and the median age at the time of the first CAD event in SHCS participants was 50 years [38].

## 4. Coronary CT Angiography (CCTA)

CCTA is a non-invasive imaging technique involving contrast-enhanced multi-slice CT for the accurate morphological assessment of the coronary arteries [102] (Figure 2). 

Importantly, and in contrast to CIMT and CAC, CCTA can be used to detect calcified and non-calcified plaques, and due to its high spatial resolution also allows for the accurate assessment of plaque composition [74,103] (Figure 3). 

CCTA has seen a tremendous development over the last two decades, both in terms of hardware advancements and improvements in image acquisition protocols. Among the latter, the introduction of prospectively electrocardiogram-triggered acquisition has contributed most substantially to lowering radiation dose exposure from 15–20 millisieverts (mSv) [104,105,106] down to 2–3 mSv [104,107,108,109,110]. The addition of sophisticated reconstruction algorithms and widespread use of latest-generation wide-coverage CT devices may now enable routine scanning in the sub-millisievert range [111]. Sensitivity of CCTA in the detection of relevant coronary stenosis is similar to standard invasive coronary angiography [112,113,114]. However, both modalities inherently lack the ability to predict hemodynamic relevance if a stenosis is found. This is of the utmost importance because the optimal treatment, i.e., the benefit of optimal medical therapy versus revascularization, depends heavily on the presence and extent of myocardial ischemia [115]. While, theoretically, measurement of pressure drops along a vessel and the derivation of the so-called fractional flow reserve constitutes an invasive technique to overcome this limitation, the application of hybrid imaging, that is, the combination and fusion of two non-invasive imaging modalities, has evolved into a robust and elegant technique for the assessment of both coronary anatomy and potential hemodynamical effects of any given lesion, hence providing the non-invasive grounds for optimized patient management decisions (Figure 4).

Compared to intravascular ultrasound, CCTA can precisely detect coronary plaque with a positive and negative predictive value >98% [18,107,116,117]. It is well recorded, however, that CCTA tends to overestimate the degree of coronary stenosis [118,119]. In the general population, among individuals without known CVD, the large scale CONFIRM study documented that the degree of non-obstructive and obstructive coronary artery stenosis and the number of vessels affected in CCTA are closely correlated with CV events and mortality rates over 3 years of follow-up [107]. CCTA improves prediction of future CV events compared to Framingham risk score [120], CAC [107,108,121] or CIMT [108,118]. 

In early studies, results regarding the presence of subclinical atherosclerosis in CCTA in HIV-positive relative to HIV-negative persons were not uniform. Some investigators recorded an increased [44,100,122,123,124] prevalence, while others noted a similar [18,43,125] or even lower prevalence in PLWH, compared to HIV-negative controls [18,103]. In a 2015 meta-analysis summarizing initial CCTA studies (1229 HIV-positive and 1029 HIV negative participants), a 3-fold higher prevalence of non-calcified coronary artery plaque on CCTA was recorded in asymptomatic HIV+ compared to HIV-negative subjects [100]. The analyzed studies typically included a small number of participants, that, in addition, were not always matched on CV risk factors [124].

Today, there are three published CCTA studies that enrolled large numbers of HIV+ and HIV-negative participants, with the following key findings: In men who had sex with men that were enrolled in the US-American multicenter AIDS cohort study (MACS), Post et al. [44] recorded a higher prevalence of any coronary plaque and non-calcified plaque in 618 HIV+ compared to 383 HIV-negative men in the baseline CCTA. In a follow-up report, with a median interval of 4.5 years between CCTAs, increased progression of coronary plaques was observed in HIV-positive compared to the HIV-negative MACS participants, but only in men with detectable HIV viremia and, thus, presumably, higher degrees of ongoing systemic inflammation, compared to persons with optimal virological control [44,126]. 

Lai et al. described the presence of subclinical atherosclerosis in CCTA of 953 HIV+ and 476 HIV-negative African American study participants from Baltimore/USA of whom more than 50% were frequent cocaine users [43]. In this study, there was a trend towards HIV infection being independently associated with non-calcified plaque, however, this association was largely explained by chronic cocaine use. 

In addition to these two studies from the US [43,44,126], we recently assessed subclinical atherosclerosis in a cross-sectional study of 428 HIV-positive participants of the Swiss HIV Cohort Study (SHCS) and 276 HIV-negative control individuals with similar Framingham risk scores. HIV-positive participants had similar degrees of non-calcified/mixed plaque and high risk [97] plaque, and, indeed, had less calcified coronary plaque, and lower coronary atherosclerosis involvement and severity scores compared to their negative controls with similar Framingham risk score [18]. These Swiss CCTA results, together with Swiss data showing similar CV incidence rates in HIV+ and HIV-persons [8], attenuate concerns about accelerated atherosclerosis in HIV, and have prompted experts to ask the question whether the prevalent notion of increased CV risk in PLWH amounts to “much ado about nothing” [127]. 

The different CCTA results in these three large studies [18,43,44] might be related to a number of features. First, CAD rates are lower in southern/central Europe compared to the US [128,129]. Second, we speculate that the low subclinical atherosclerosis prevalence in Swiss PLWH might further be attributable to high rates of successful treatment in the setting of modern ART regimens, regular follow-up, and declining smoking rates in recent years [9,130].

As regards the duration of ART, the Swiss study did not find any association with any type of plaque. This was in line with findings from Lai et al. in the group of non-cocaine users and contrary to MACS [18,43,44]. Concerning the use of antiretroviral agents and subclinical atherosclerosis, the MACS investigators observed no clear association between different ART drugs and any form of coronary artery plaque on CCTA [131]. In contrast, the Swiss study identified an association with exposure to abacavir and an increased prevalence of non-calcified/mixed plaque [132]. This observation seems important, as it may afford a mechanism by which abacavir increases CV risk, i.e., the promotion of atherosclerosis, in addition to putative deleterious effects of abacavir on platelet reactivity [133,134].

Consistent with the MACS findings, the Swiss patients with a low nadir CD4+ cell count had more non-calcified plaque [18,44] and more coronary stenosis greater than 50% [44]. These findings suggest that advanced immunosuppression may contribute to subclinical atherosclerosis in PLWH [100].

In summary, coronary plaque in young persons typically is non-calcified [80,99,100]. CCTA (but not CAC) can detect non-calcified plaque [74] and predicts CVD better than CAC [107] or CIMT [118]. Therefore, and because the median age of most populations of PLWH remains below 50 years today, CCTA is emerging as the preferred research-setting imaging tool to detect subclinical atherosclerosis. There remains a need for additional studies to examine the exact role that CCTA may play in clinical HIV practice.

## 5. Magnetic Resonance Rmaging (MRI)

Based on CIMT being strongly predictive of CV events [135], multiple studies have reported the use of MRI rather than ultrasound in assessing CVD in PLWH [103,136,137,138,139]. Such studies mainly used black-blood MRI (BBMRI), an advanced imaging technique relying on nulling of the MR signal from the vascular lumen while retaining signal from the vascular wall, allowing better visualization of the wall thickness [140,141]. Multiple variations of the BBMRI technique have already been used in extracranial (e.g., aorta, carotid and coronary arteries) as well as intracranial vessel wall imaging [140,141,142,143,144,145,146].

In age-related atherosclerosis, BBMRI is generally used to assess the vulnerability characteristics of eccentric, lipid-rich plaques (e.g., lipid core, fibrous cap, hemorrhage). In HIV vasculopathy on the other hand, where intimal smooth muscle hypertrophy occurs in a symmetrical circumferential manner [32,33], BBMRI is more commonly used to assess diffuse wall thickening. This is why the majority of vessel wall imaging studies in HIV have concentrated on measuring the vessel wall thickness (VWT) akin to the measurement of CIMT using ultrasound [103,138,147,148]. In a group of treated HIV+ subjects with low measurable CVD risk, HIV-status was significantly associated with increased wall/outer-wall ratio (W/OW), an index of vascular thickening, after adjusting for age [139]. In another study, increased carotid artery wall thickness was noted in HIV+ subjects on chronic ART compared to controls with similar CVD risk, [137]. Interestingly, while in one study [139], no correlation of CVD with the type of ART was identified, another study [137] found that a longer duration of protease inhibitor therapy was associated with greater wall thickness. Using MRI to assess coronary rather than carotid arteries, subclinical CAD has been documented in ART-treated subjects, with HIV infection being independently associated with increased proximal right coronary artery wall thickness [103].

The connection between HIV-associated inflammation and atherosclerosis burden in HIV infection was also evaluated in a group of HIV+ subjects (treated and untreated) in comparison to a group of demographically similar controls. As expected, HIV-1 viral burden was found to be associated with higher serum levels of the chemokines monocyte chemoattractant protein-1/CC-chemokinligand 2 (MCP-1/CCL2). At the same time, HIV infection correlated with atherosclerosis burden, mainly detectable as increased vessel wall area and thickness in the thoracic aorta [148].

BBMRI has also been important in delineating that traditional CVD risk factors, which seem to disproportionately burden PLWH, need to be controlled. In one study evaluating asymptomatic, young-to-middle-aged African-Americans with and without HIV infection and cocaine use, only total cholesterol (and not HIV status) was significantly associated with the presence of a lipid core in carotid plaques [136]. In another more recent study, carotid VWT was increased in treated PLWH relative to controls with the 10-year ASCVD (atherosclerotic cardiovascular disease) risk score being the only variable significantly associated with VWT whereas HIV status was not. The authors concluded that traditional CVD risk factors in PLWH are adequately captured in the ASCVD risk score which was closely associated with subclinical carotid disease [138].

Over the last few years, there have been significant advances made in MR imaging of the vessel wall, with the ability nowadays of assessing atherosclerotic involvement of the intracranial arteries. To our knowledge, such high-resolution techniques have not yet been applied in the evaluation of PLWH [146,149,150,151,152,153]. At this point, BBMRI of the vessel wall remains a research application rather than a clinically used technique in assessing CVD in PLWH (Figure 5). Whether this would eventually change in the future, as PLWH get older, remains unclear. 

## 6. 18F-Fluorodeoxyglucose Positron Emission Tomography (FDG PET)

The most common application of FDG PET imaging in the clinic is the baseline assessment and follow-up of neoplastic diseases. Applications in imaging infectious and inflammatory diseases using FDG PET, however, have been increasing in the last decade.

One such use is the assessment of vasculitis seen as increased glucose metabolism of the vessel wall, reflecting inflammatory changes [154] (Figure 6). 

A similar approach has been used by multiple groups to assess HIV-vasculopathy using FDG PET [155,156,157,158,159,160,161,162]. A common presumption is that PLWH will demonstrate increased vessel wall FDG uptake reflecting premature aging and atherosclerotic changes. In fact, in one paper assessing only HIV+ patients, a relationship was found between arterial inflammation on FDG PET and features of high-risk coronary atherosclerotic disease including number of low attenuation plaques and plaque vulnerability characteristics [160].

The results from studies utilizing FDG PET to assess HIV-vasculopathy, however, have been rather inconsistent. In one paper [162], arterial inflammation in the aorta (measured as aortic target to background ratio (TBR)) was higher in the HIV+ group compared to the non-HIV control group matched for CVD, even after adjusting for traditional CV risk factors [162]. More recently, HIV was found to be associated with 0.16 higher aortic TBR compared to controls, independent of traditional risk factors [155]. On the other hand, in one paper assessing virologically controlled HIV+ persons with no known CVD, there was no evidence of increased arterial inflammation on FDG PET compared to healthy volunteers [159]. In a more recent paper looking at a larger group of patients, there was only a marginally higher TBR in PLWH, with significant overlap of TBR values with HIV-negative controls [156]. The discrepancy in the results of those studies could be related to variations in study populations, treatment status of patients and the presence or absence of co-morbid CVD. Another element of variability could also be related to the imaging technique and analysis methods.

Another use for FDG PET is in longitudinally assessing PLWH for changes in arterial inflammation over time. To our knowledge, only one such longitudinal study has been reported, with twelve ART naïve subjects imaged at baseline and again six months after ART initiation. Interestingly, there were no significant changes in TBR between baseline and follow-up scans [157]. This could be due to short interval imaging after ART initiation. Another paper assessing PLWH treated with statins versus placebo found no significant change in arterial inflammation as measured by FDG PET despite the fact that statin therapy reduced non-calcified plaque volume and high-risk coronary plaque features on CCTA [158].

At this point, using FDG PET in PLWH is more likely to be useful in evaluating co-morbidities, specific organ involvement or assessment of central nervous system involvement [163,164,165,166,167]. The role of FDG PET in assessing CVD in individual subjects on the other hand remains uncertain.

## 7. Outlook

Consistent with Swiss data [8], a nationwide cohort study in Denmark found that after the exclusion of participants with risk factors including illicit drug use or hepatitis B or C co-infections, the mortality rate was similar in HIV-positive and HIV-negative persons [168]. In general, persons with well-controlled HIV infection and a healthy lifestyle appear to have similar survival rates today as do HIV-negative persons, with potentially increased death rates mainly ascribed to traditional risk factors [168,169]. Thus, for efficient primary CAD prevention in PLWH, our aim should be the optimization of traditional risk factor management, early initiation of ART therapy, and regular patient follow-up.

Few studies have simultaneously applied more than one cardiac imaging modality (CIMT and CAC [170], CCTA and MRI [103]) in the same patients, but did not correlate CIMT results with CAC or MRI with CCTA results. Hsue and colleagues found that with or without detectable CAC, PLWH had higher CIMT than HIV-negative controls [171]. Interestingly, among those without detectable CAC, a third of PLWH but no controls had increased CIMT.

CCTA is emerging as an accurate and reliable non-invasive imaging modality for detection of subclinical atherosclerosis in PLWH, especially in persons <50 years who typically have non-calcified or partially calcified plaque. However, in asymptomatic PLWH today, cardiac imaging is not yet indicated to assess coronary atherosclerosis. The exception might be asymptomatic persons with diabetes [172]. Studies that compared coronary imaging to optimization of medical treatment in asymptomatic patients have not yet shown any clinical benefit, but a number of relevant studies are ongoing. Particularly in patients with atypical symptoms suggestive of myocardial ischemia, or in patients with suspected acute coronary syndrome but without ST-segment elevations on EKG, CCTA is now increasingly well established as a valuable diagnostic tool to rule out CAD in the emergency room [173,174,175], because of its non-invasiveness compared to coronary angiography and its extremely high negative predictive value [176].

## Figures and Tables

**Figure 1 jcm-08-01125-f001:**
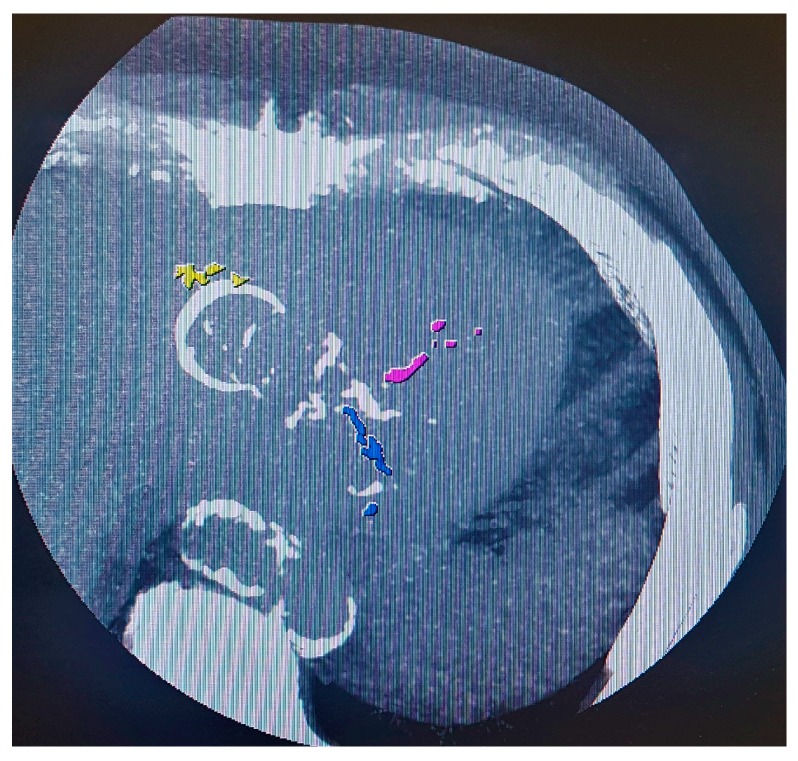
Coronary artery calcium scoring in an asymptomatic 43-year old HIV-positive male patient. Maximum intensity projection depicts extensive calcifications in the left anterior descending artery (purple), in the left circumflex artery (blue), and in the right coronary artery (yellow). The total Agatston score was 1031, classifying this patient as at high risk for future CV events and prompting lifestyle interventions and the initiation of a statin.

**Figure 2 jcm-08-01125-f002:**
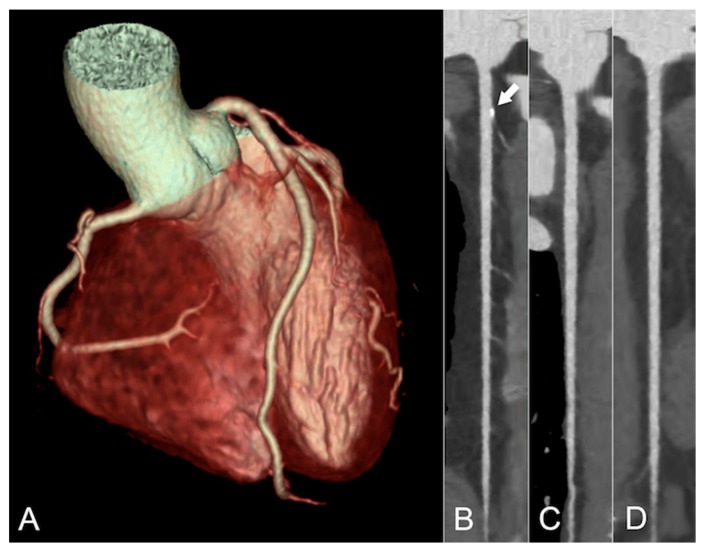
Coronary computed tomography angiography of a 48-year old HIV+ female patient with typical angina. The patient had a positive family history for cardiovascular events and was an active smoker, putting her at intermediate risk for having coronary artery disease. Volume rendering (**A**) of the image datasets depicts normal coronary anatomy, while the curved multiplanar reformats of the left anterior descending (**B**), the left circumflex (**C**), and the right coronary (**D**) arteries reveal only minimal and non-obstructive coronary atherosclerosis in the proximal left anterior descending (white arrow). Coronary artery disease was, therefore, excluded as a cause for her symptoms. BMI 24.2 kg/m^2^. Radiation dose exposure 0.57 mSv. Contrast volume 40 mL.

**Figure 3 jcm-08-01125-f003:**
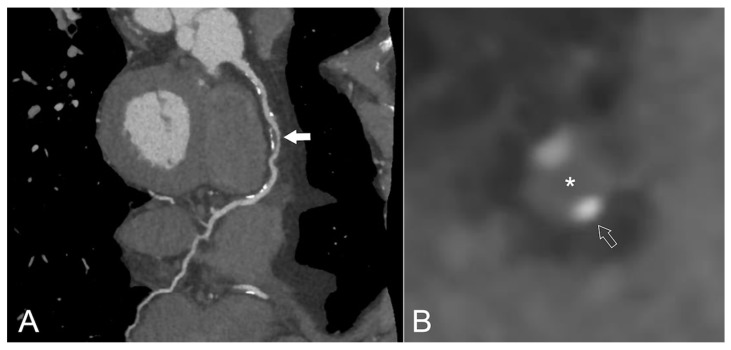
Non-invasive evaluation of an asymptomatic 65-year old HIV+ male patient. The patient had a history of treated arterial hypertension, was an active smoker and suffered from peripheral artery disease with the latter prompting further cardiovascular work-up despite a normal stress-electrocardiogram. Multiplanar curved reconstruction of the coronary computed tomography angiography dataset (**A**) reveals multiple calcified but non-obstructive lesions and 70–90% stenosis (white filled arrow) in the mid-right coronary artery. This obstructive lesion (**B**, cross sectional view) exhibits several morphological high-risk features typical for an event-prone lesion, such as positive remodeling and a low-attenuation core (*) with calcifications only at the edge of the lesion (white empty arrow), constituting the so-called napkin-ring sign.

**Figure 4 jcm-08-01125-f004:**
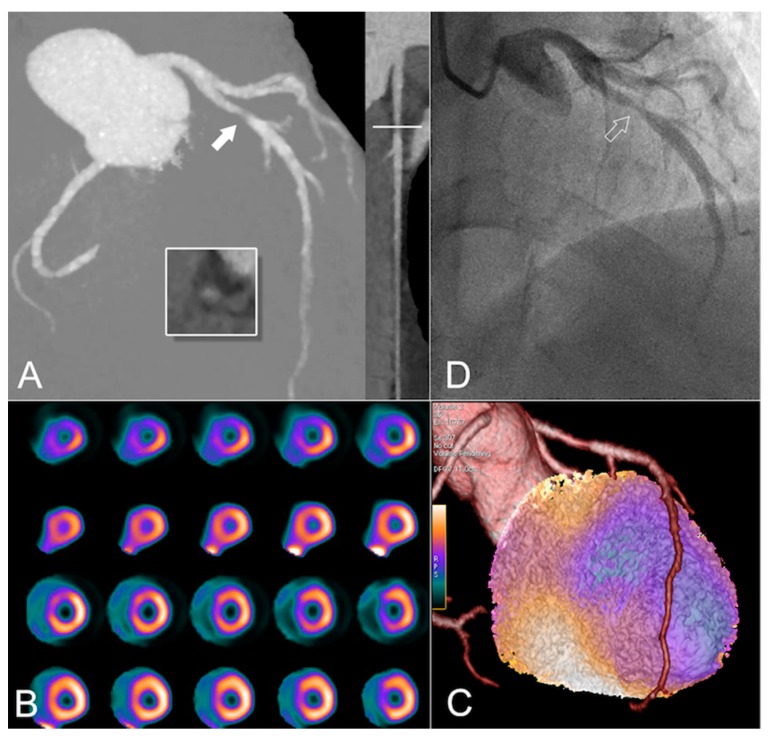
Evaluation of a 49-year old HIV+ male patient with typical angina. The patient had no cardiovascular risk factors and he was referred for exclusion of coronary artery disease with a calculated pre-test probability of 69%. Total cholesterol was 3.6 mmol/L (<5.0), HDL-cholesterol 1.45 mmol/L (>1.0), and LDL-cholesterol 1.8 mmol/L (<3.0). Maximum intensity projection and multiplanar curved reconstruction of the coronary computed tomography angiography (CCTA) datasets (**A**) depict a 70–90% stenosis (white filled arrow) in the proximal left anterior descending artery as shown in the cross-section (inlet). As per clinical routine and in accordance with current guidelines, lesions are primarily graded visually with regard to maximal percent diameter stenosis in our institution. In this particular patient, the qualitative evaluation was complemented by quantitative measurements (using version 2 of CardIQ Xpress/GE Healthcare), because the lesion in the LAD was non-calcified, allowing for more accurate lumen delineation due to the lack of blooming artifacts. The quantitative analysis resulted in a diameter stenosis of 75%. Myocardial perfusion single-photon-computed-emission-tomography (SPECT, (**B**)) confirmed hemodynamical relevance of the lesion, revealing a large perfusion defect during stress (top rows) with reversibility during rest (bottom rows), constituting ischemia in the anteroseptal wall of the left ventricular myocardium. Hybrid CCTA/SPECT imaging (**C**) clearly demonstrates a large area of ischemia in the myocardium subtended by the left anterior descending artery. Finally, obstructive coronary artery disease (white empty arrow) was confirmed during invasive coronary angiography (**D**), and the lesion was treated with a drug-eluting stent.

**Figure 5 jcm-08-01125-f005:**
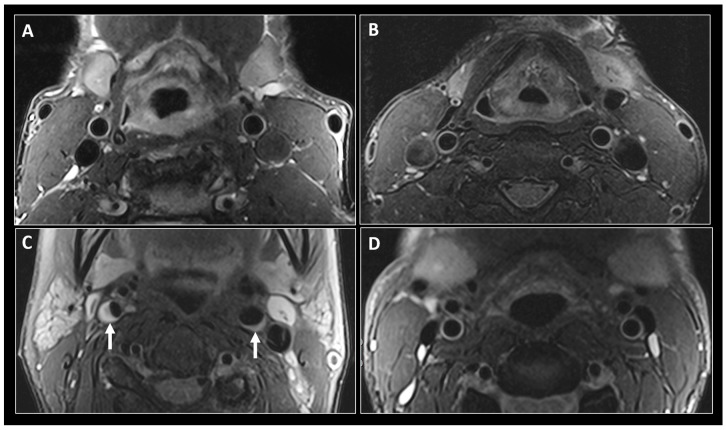
Black-blood MR imaging of the carotid arteries. Fat saturated T2-weighted black-blood MR images at the level of the common carotid arteries in a 56-year-old HIV+ man (**A**) and a 47-year-old HIV− man (**B**). Similar imaging technique at the level of the internal carotid arteries (ICAs) in a 56-year-old HIV+ woman (**C**) shows narrowing of the vascular lumen bilaterally by a plaque (small arrows), more significant on the right side. (**D**) shows similar imaging at the level of ICAs in a 47-year-old HIV negative man with no evidence of atherosclerosis.

**Figure 6 jcm-08-01125-f006:**
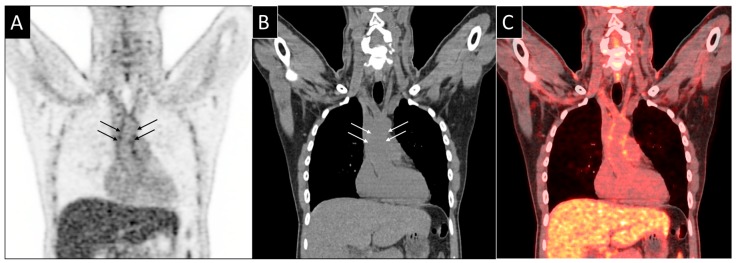
FDG-PET imaging of the major vessels. (**A**) Coronal FDG PET images, (**B**) coronal CT scan images and (**C**) fused PET CT scans show subtle FDG uptake in the vascular wall of the ascending aorta (arrows).

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
