# Peer review of "Subclinical Atherosclerosis Imaging in People Living with HIV"

_jcm, 2019, doi:10.3390/jcm8081125_

Round 1
Reviewer 1 Report
This review paper is very well written, provides recent highlights regarding the incident of CVDs in people living with HIV.
I recommend for publication after minor correction.
Page 2, line 67-70 "Several serum biomarkers that are associated with CV risk or overall mortality may be elevated in PLWH compared to HIV-negative persons, including high sensitivity C-reactive protein, D-dimer, interleukin-6, tumor necrosis factor, soluble CD14 and CD163"
These sentence were written from reference 24-26. In these papers, there is also mention of other serum biomarkers associated with CV which is missing in this review paper. Please include them. Also, it would be better if authors could add few information of why these serum biomarker are important. Example: high level of ICAM-1 facilitate the adhesion of circulating cell to endothelial followed by transmigration, leading to vascular inflammation.
Page 2, line 90-93 "According to the results of a 2009 meta-analysis that analyzed 6 cross-sectional, 7 case-control and 13 cohort studies (5456 HIV positive and 3600 HIV negative patients) PLWH tend to show a greater thickness in CIMT (0,04 mm thicker; 95% confidence interval: 0,02- 0,06 mm, p< 0.001), compared to HIV-negative controls"
It is mention that the thickness of carotid artery intima-media increase but it is not clear why the thickness increase. It would be better to mention a sentence about it [proliferation and migration of vascular smooth muscle cell, migration of circulating cell (monocyte/macrophage etc) to intima].
Figure 4, figure legend. How was 70-90% stenosis calculated?
Figure 5, B and D are image from HIV negative and A and C are image from HIV positive, However, only 5C has narrowing of blood vessel due to plaque. Why did author choose image A with no narrowing of blood vessel??
Full form for some abbreviation is missing. For some abbreviation, the full form is mention later in the manuscript. Full form must be mentioned when it is abbreviated first in the paper.
Author Response
1.) Page 2, line 67-70 "Several serum biomarkers that are associated with CV risk or overall mortality may be elevated in PLWH compared to HIV-negative persons, including high sensitivity C-reactive protein, D-dimer, interleukin-6, tumor necrosis factor, soluble CD14 and CD163". These sentence were written from reference 24-26. In these papers, there is also mention of other serum biomarkers associated with CV which is missing in this review paper. Please include them.
RESPONSE:We agree this is an important point. We have now included additional discussion of important biomarkers in the revised manuscript, as follows (page 2, lines 72-77). Please note, however, that our manuscript is a review of cardiac imaging and not of cardiac biomarkers, therefore we suggest to limit the discussion of biomarkers to the current extent.
Several serum biomarkers that are associated with CV risk or overall mortality may be elevated in PLWH compared to HIV -negative persons, including biomarkers of systemic inflammation (high sensitivity C-reactive protein, interleukin-6, soluble tumor necrosis factor-α receptor 1) 24–27, coagulation (D-dimer, fibrinogen) 24–27, monocyte activation (lipopolysaccharide, soluble CD14, CD163, CC-chemokine ligand 2) 24,28–31and endothelial dysfunction (intercellular adhesion molecule 1) 24.
2.) Also, it would be better if authors could add few information of why these serum biomarker are important. Example: high level of ICAM-1 facilitate the adhesion of circulating cell to endothelial followed by transmigration, leading to vascular inflammation.
Page 2, line 90-93 "According to the results of a 2009 meta-analysis that analyzed 6 cross-sectional, 7 case-control and 13 cohort studies (5456 HIV positive and 3600 HIV negative patients) PLWH tend to show a greater thickness in CIMT (0,04 mm thicker; 95% confidence interval: 0,02- 0,06 mm,p< 0.001), compared to HIV-negative controls".
It is mention that the thickness of carotid artery intima-media increase but it is not clear why the thickness increase. It would be better to mention a sentence about it [proliferation and migration of vascular smooth muscle cell, migration of circulating cell (monocyte/macrophage etc) to intima].
RESPONSE: We agree and have added a section about the pathophysiology of atherosclerosis in HIV+ in the revised manuscript (page 2, lines 77-83).
Although the exact mechanisms underlying the development of atherosclerosis in PLWH have not been conclusively determined, smooth muscle cell proliferation may play an important role 32. Moreover, endothelial dysfunction due to adhesion molecules may lead to adhesion of circulating leukocytes, transmigration of monocytes/macrophages to the intima and subsequently to vascular inflammation 20,32,33. In combination with procoagulatory mechanisms in the setting of increased concentrations of D-dimer and coagulation factors, and in presence of other metabolic disorders (e.g. dyslipidemia), the progression of atherosclerosis might be enhanced in PLWH 20,32.
3.) Figure 4, figure legend. How was 70-90% stenosis calculated?
RESPONSE: Thank you for this valuable question. We have now added the following information to the figure legend (page 7, line 236-241):
As per clinical routine and in accordance with current guidelines, lesions are primarily graded visually with regard to maximal percent diameter stenosis in our institution. In this particular patient, the qualitative evaluation was complemented by quantitative measurements (using version 2 of CardIQ Xpress (GE Healthcare)), because the lesion in the LAD was non-calcified, allowing for more accurate lumen delineation due to the lack of blooming artifacts. The quantitative analysis resulted in a diameter stenosis of 75%.
4.) Figure 5, B and D are image from HIV negative and A and C are image from HIV positive, However, only 5C has narrowing of blood vessel due to plaque. Why did author choose image A with no narrowing of blood vessel??
RESPONSE: Thank you for this useful relevant question. Contrary to what possibly might be expected, not all studies show more pronounced atherosclerosis in HIV+ compared to HIV-negative persons. The selected images were chosen for didactic purposes, and irrespective of any notion of accelerated atherosclerosis in HIV, because they are of high quality and because they convincingly show, side by side, what a normal carotid looks like and what an atherosclerotic carotid looks like on MRI.
5.) Full form for some abbreviation is missing. For some abbreviation, the full form is mention later in the manuscript. Full form must be mentioned when it is abbreviated first in the paper.
RESPONSE:Done.
Reviewer 2 Report
The manuscript entitled “Subclinical Atherosclerosis Imaging in People Living with HIV” by Isabella C. Schoepf and colleagues aims to establish an early and non-invasive imaging method to detect asymptomatic (subclinical) atherosclerosis in people living with HIV. This review is well written in general and has included up-to-date references in the field. Below are suggestions to strengthen the study further.
1. All figures appear to be case images assessed by one imaging method. To claim one method is better than another, comparative images between different imaging methods or same imaging method for different stages of the disease should be provided.
2. Other factors besides HIV including cardiovascular history and lifestyle should also be discussed when interpreting imaging data from clinical cases
3. The potential mechanisms of HIV and atherosclerosis need to be discussed and incorporated into the significance of the early and non-invasive imaging methods
Author Response
1.) All figures appear to be case images assessed by one imaging method. To claim one method is better than another, comparative images between different imaging methods or same imaging method for different stages of the disease should be provided.
RESPONSE: This is an important comment and the author correctly points out that studies that compare atherosclerotic findings using different techniques (e.g. CIMT vs CCTA vs MRI) would be of high interest. We are aware of only few comparative studies in PLWH and controls (Most studies have employed either CIMT or CCTA or MRI, but not simultaneously all techniques in the same patient). We have added the following section to the revised manuscript (page 12, lines 422-426)
Few studies have simultaneously applied more than one cardiac imaging modality (CIMT and CAC 172, CCTA and MRI103) in the same patients, but did not correlate CIMT results with CAC or MRI with CCTA results. Hsue and colleagues found that with or without detectable CAC, PLWH had higher CIMT than HIV-negative controls173. Interestingly, among those without detectable CAC, a third of PLWH but no controls had increased CIMT.
2.) Other factors besides HIV including cardiovascular history and lifestyle should also be discussed when interpreting imaging data from clinical cases.
RESPONSE: We agree with the reviewer that the interpretation of cardiac imaging results must integrate all available imaging and non-imaging information. We have now added the most relevant clinical information to the figure legends.
Figure 2.Coronary computed tomography angiography of a 48-year old HIV+ female patient with typical angina. The patient had a positive family history for cardiovascular events and was an active smoker, putting her at intermediate risk for having coronary artery disease.
Figure 3.Non-invasive evaluation of an asymptomatic 65-year old HIV+ male patient.The patient had a history of treated arterial hypertension, was an active smoker and suffered from peripheral artery disease with the latter prompting further cardiovascular work-up despite a normal stress-electrocardiogram.
Figure 4.Evaluation of a 49-year old HIV+ male patient with typical angina.The patient had no cardiovascular risk factors and he was referred for exclusion of coronary artery disease with a calculated pre-test probability of 69%. Total cholesterol was 3.6 mmol/l (<5.0), hdl-cholesterol="" 1.45="" l="">1.0), and LDL-cholesterol 1.8 mmol/l (<3.0).< span="">
3.) The potential mechanisms of HIV and atherosclerosis need to be discussed and incorporated into the significance of the early and non-invasive imaging methods.
RESPONSE: We agree. Please refer to our response to reviewer 1 who made a similar comment in their comment 2.
